# The mass of the $\pi^-$

M. Daum[1] and D. Gotta[2*]

**1** Paul Scherrer Institut, 5232 Villigen PSI, Switzerland
**2** Institut für Kernphysik, Forschungszentrum Jülich, 52425 Jülich, Germany
\* d.gotta@fz-juelich.de

February 22, 2021

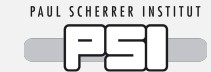

## Abstract

**The most precise values of the mass of the negatively charged pion have been determined from several measurements of X-ray wavelengths for transitions in pionic atoms at PSI. The Particle Data Group gives the average $m_{\pi^-} = $ (139.570 61 $\pm$ 0.000 24) MeV/c$^2$.**

## 10.1 Introduction

The most accurate determination of the mass of the negatively charged pion, $m_{\pi^-}$, is obtained from measurements of X-ray transition energies in pionic atoms. X-rays stem from a de-excitation cascade after capture into high-lying atomic states of a nucleus $N_Z^A$ with mass number $A$ and charge $Z$.

The atomic binding energies $E_{nl}$ are directly related to the reduced mass $\mu$ of the $\pi N_Z^A$ system. The relativistic description of $E_{nl}$ is given for spin 0 particles by [1]

$$E_{nl} = \frac{-\mu c^2}{2} \left(\frac{Z\alpha}{n}\right)^2 \left[1 + \left(\frac{Z\alpha}{n}\right)^2 \left(\frac{n}{l+1/2} - \frac{3}{4}\right)\right] + \mathcal{O}\left[(Z\alpha)^6\right]. \qquad (10.1)$$

Here, $n$ and $l$ are the principal and angular momentum quantum numbers of the atomic level, respectively, and $\alpha$ is the fine structure constant. The leading term of $\mathcal{O}\left[(Z\alpha)^2\right]$ coincides with the well-known Bohr formula. (10.1) holds for $Z \lesssim 1/(2\alpha) = 68$.

For high-precision experiments, further contributions to $E_{nl}$, not included in (10.1), must be considered. Most important are QED effects, i.e. vacuum polarization, relativistic recoil ($\mathcal{O}\left[(Z\alpha)^4\right]$), as well as hyperfine and strong-interaction shifts. Recent QED calculations achieve an accuracy of $\leq \pm 1\,\text{meV}$ for pure electromagnetic transition energies [2].

## 10.2 Measurements at PSI

New measurements began following discussions of muon neutrino mass limits, aiming at a precision of about 1 ppm for the mass of the $\pi^-$. The three most recent and precise determinations of $m_{\pi^-}$ [3] were performed at PSI, using the high pion fluxes available there. The X-ray transition energies $E_X$ are obtained via the measurement of the angle of diffraction, the Bragg angle $\Theta_B$, with crystal spectrometers by using Bragg's law $n\lambda = 2d \cdot \sin\Theta_B$, where $n$ is the order of reflection, $\lambda = h/E_X$ the X-ray's wave length, $h$ Planck's constant, and $d$ the lattice constant of the corresponding crystal planes.

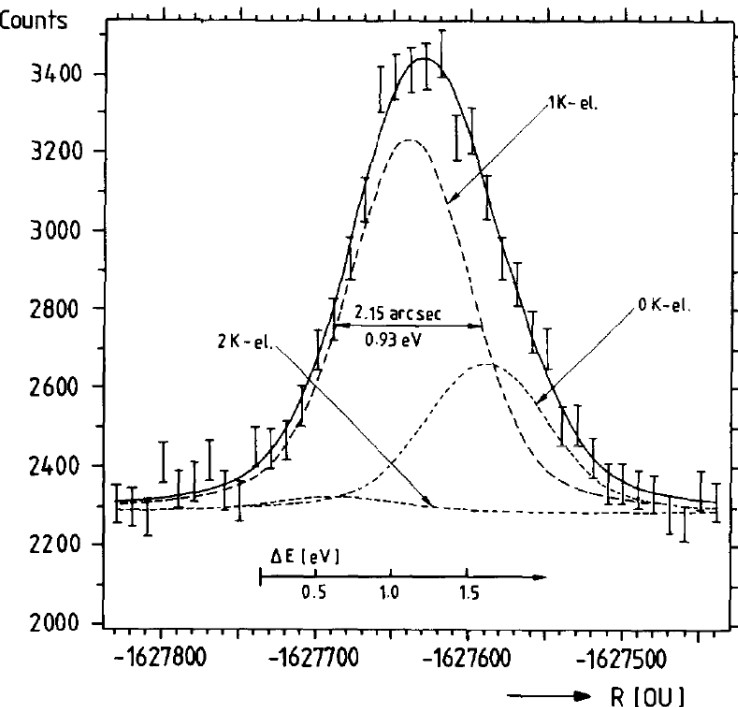

Figure 10.1:  Bragg reflection of the $(4f-3d)$ transition in pionic $^{24}$Mg measured with a (110) quartz crystal in third order of diffraction; x-axis: R is the interferometer read-out in optical units (OU). The fit function is marked by the solid line; it is the sum of three individual peaks corresponding to the cases of having two, one or zero K-electrons present during the pionic transition. The line shapes of the different peaks are obtained by folding the instrumental response function with the natural line width of the transition.

In the first of these experiments, a DuMond crystal spectrometer was used to measure the $\pi$Mg$(4f-3d)$ transition at 26.9 keV in a solid magnesium target [4, 5]. Energy calibration and experimental resolution were provided by the 25.7 keV $\gamma$ line from $^{161}$Tb decay. The observed line width, however, was larger than the instrumental resolution of 0.93 eV (Figure 10.1). This was attributed to the occurrence of different populations of the electronic K shell and, consequently, different screenings of the nuclear charge. Based on a measurement of the intensity balance of the sum of the $(nf-3d)$ transitions to the $(3d-2p)$ line, which yielded a K electron shell population of $(0.44 \pm 0.30)$, it was originally assumed that the strongest component in the spectrum corresponds to one K-shell electron. The corresponding result for the pion mass (solution A) is given in Table 10.1 - entry 1986.

Later, this result came into strong disagreement with the continuously improved precision measurements of the muon momentum $p_{\mu^+}$ from pion decay at rest $\pi^+ \rightarrow \mu^+\nu_\mu$ [9–11]. The lower limit thus derived for $m_{\pi^+}$ was 3.5 standard deviations higher than the world average for $m_{\pi^-}$ as obtained from pionic magnesium. In addition, the squared muon neutrino mass determined from $p_{\mu^+}$ and $m_{\pi^-}$ then became negative by 6 standard deviations [10, 11].

A re-assessment of the $\pi^-$Mg$(4f-3d)$ line shape experiment led to the conclusion that when interpreting the strongest component in Figure 10.1 as the two K-electron contribution [6], the above-mentioned discrepancy in the $m_{\pi^+}$ results is removed. The alternative value for $m_{\pi^-}$ (solution B) is given in Table 10.1 - entry 1994. This is in

| year | method | $m_{\pi^-}$ / MeV/c$^2$ | reference |
|------|--------|------------------------|-----------|
| 1986 | $\pi$Mg$(4f-3d)/^{161}$Tb $\gamma$ (A) | 139.568 71 $\pm$ 0.000 53 | [4, 5] |
| 1994 | $\pi$Mg$(4f-3d)/^{161}$Tb $\gamma$ (B) | 139.569 95 $\pm$ 0.000 37 | [6] |
| 1998 | $\pi$N$(5g-4f)/$Cu K$\alpha$ | 139.570 71 $\pm$ 0.000 53 | [7] |
| 2016 | $\pi$N$(5g-4f)/\mu$O$(5g-4f)$ | 139.570 77 $\pm$ 0.000 18 | [8] |
| 2018 | $\pi^-$ PDG average | 139.570 61 $\pm$ 0.000 23 | [3] |

Table 10.1:   Recent results for the mass of the negatively charged pion. The PDG derived an average from the entries 1994, 1998, and 2016. The uncertainty includes a scale factor of 1.6. Earlier measurements have been omitted as they may have incorrect K-shell screening corrections [3].

line with the discussion on the ionization state during the de-excitation cascade, which assumes a continuous refilling of electrons for metals [12].

In view of the importance of the questions involved, a new measurement of the $\pi^-$ mass was undertaken [7]. The increased pion flux resulting from the larger proton current in the PSI cyclotron allowed the use of the cyclotron trap [13, 14], gas targets of about 1 bar pressure (NTP), and a Johann-type crystal spectrometer. The big advantage of gaseous targets is that K-electron contamination is expected to be small [12].

The $(5g-4f)$ transition in pionic nitrogen is an ideal candidate. With an energy of 4.055 keV, the reflectivity of silicon Bragg crystals in second order and the efficiency of X-ray detectors are close to optimum. The copper K$\alpha_1$ fluorescence line of 8.048 keV provides the energy calibration at practically the same Bragg angle when measured in fourth order [7]. As in the $\pi$Mg case, different electron screening contributions would be apparent as distortions of the line shape. The energy shift due to one (two) K electron(s) is $-456\,(-814)$ meV, while the spectrometer resolution is about 450 meV. The natural line width of 8 meV is negligibly small, and strong-interaction effects in the $4f$ level can be estimated sufficiently accurate. The mass value derived from the $\pi$N$(5g-4f)$ transition (Figure 10.2) is in agreement both with solution B of the $\pi$Mg experiment [6] and the results deduced from $\pi^+$-decay [10, 11] (Table 10.1 - entry 1998).

In a second experiment, the two shortcomings of the Cu calibration were avoided: (i) Spectra of fluorescence X-rays always include satellite lines from multiple ionization depending on details of the excitation conditions. Therefore, measured energies may slightly deviate from published reference values. (ii) Measuring in different orders of reflection requires substantial corrections to the Bragg angle resulting in additional uncertainties [7].

A comparison of X-ray transition energies shows a near coincidence for $\mu$O and $\pi$N. The muonic line provides an accurate calibration due to the precise knowledge of the muon mass to 23 ppb [3, 15, 16]. Choosing again the $(5g-4f)$ lines for both atoms and using a O$_2$/N$_2$ gas mixture allows a simultaneous measurement in the same order of reflection without any manipulation of the set-up [8] (Figure 10.3). The result of this measurement agrees well with the previous $\pi$N measurement [7] (Table 10.1 - entry 2016).

The measured $\pi$N and $\mu$O line widths are $\approx 800$ meV, much larger than the spectrometer resolution. The increase of the widths is due to Doppler broadening from Coulomb explosion, a recoil effect appearing in molecules [17], and, in contrast to $\pi$Mg, not to any electron screening. The analysis of the $\pi$N$(5g-4f)$ line shape provides an upper limit for the K-electron contamination of $10^{-6}$, which is much less than the 10% predicted by cascade calculations [18], but corroborates the results from experiments measuring the density dependence of X-ray yields [19]. Measuring the fine-structure splitting generated by the angular momentum dependence in pionic atoms, gives the best available test of the

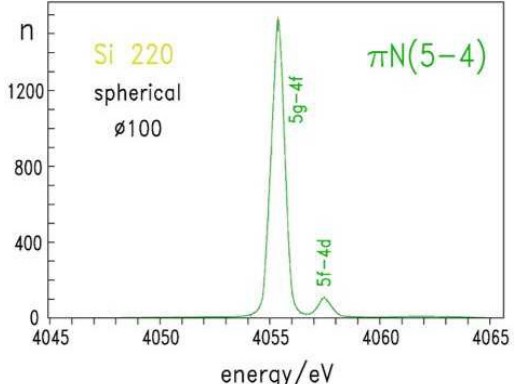

Figure 10.2: $\pi$N$(5-4)$ complex measured with a spherically bent Si(110) crystal in $2^{nd}$ order. The pion mass is determined from the energy of the $\pi$N$(5g-4f)$ transition (adapted from [7]).

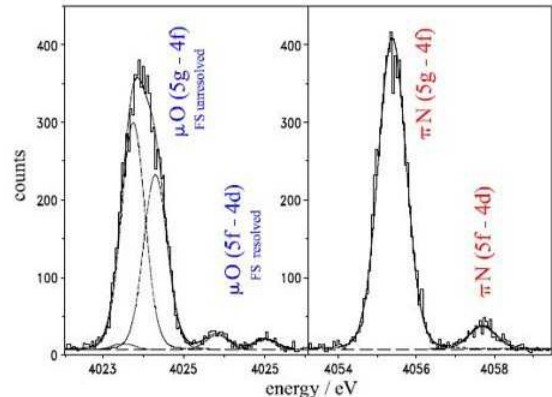

Figure 10.3: $\pi$N and $\mu$O $(5g-4f)$ transitions from the simultaneous measurement with an $O_2/N_2$ (10%/90%) gas mixture at 1.4 bar pressure (adapted from [8]).

Klein-Gordon equation, (10.1). The recent $\pi$N$(5-4)$ measurement (Figure 10.3) achieves an accuracy of 0.4% for the fine-structure splitting [7], which improves earlier tests [20,21] by one order of magnitude.

In conclusion, the present study demonstrates the potential of crystal spectroscopy with bent crystals in the field of exotic atoms. As an application, X-rays of hydrogen-like pionic atoms can be used to provide calibration standards in the few keV range, where suitable radioactive sources are not available [22]. The accuracy of such standards is given by the present uncertainty of the pion mass [2].

Facing the fact that pion beams at PSI provide a flux of about $10^9$/s, the use of double-flat crystal spectrometers may be considered allowing for absolute angle calibrations choosing specific narrow hydrogen-like pionic transitions not affected by Coulomb explosion, e. g. from pionic neon. A precision for the pion mass determination of the order of 0.5 ppm would be feasible. A method based on laser spectroscopy of metastable pionic helium, if successfully applied, could further improve significantly on the accuracy for the $\pi^-$ mass [23–25].

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
