# Peer review of "The mass of the $\pi^-$"

_SciPost Physics_

## Round 1 · Referee Report · Claude Petitjean (Referee 1) · 2021-2-17

Strengths
1- comprehensive review of the experiments that lead to a precise determination of the pion mass 2- all important papers about pion mass measurements and related objects are well referenced 3- the history of original discrepancies with other observations (p_mu), and how this was eventually resolved, is nicely and well reported 4- the most important measured spectra and the final results of pion mass values are well documented
Weaknesses
1- some accuracies given should be better explained, see requested changes 2- a general conclusion at the paper end is missing and might be beneficial
Report
Requested changes
1- introduction: the section about eq (10.1) should somehow be referenced 2- end of introduction (line 19): "<=+-1meV" - please explain of what 3-first sentence chapter 10.2 (line 22): "precision of about 1 ppm" please explain of what 4- page 3 bottom (line 84): "accuracy of 0.4%" please explain of what

Author: Detlev Gotta on 2021-02-23 [id 1261]
(in reply to Report 1 by Claude Petitjean on 2021-02-17)The changes suggested by the referee have been included. A new version is produced. The corresponding file of the new version is attached: piM_corr_2021_02_21.pdf
Changes in the text are as following (all line numbers refer to the new version) line number 13 reference related to eq. (10.1) according to referee remark 1 20-21 supplement according to referee remark 2 24 supplement according to referee remark 3 89 supplement according to referee remark 4
91-102 In view of the remark "Weakness" we added two short paragraphs as general conclusion which include more additional references.
Attachment:
piM_corr_2021_02_21.pdf

---

## Round 1 · Referee Report · Adrian Signer (Referee 2) · 2021-2-26

Report
opportunity to review an earlier draft of the article and were in
communication with the authors before the submission. All our comments
and suggestions have been taken into account. Hence, we think the
paper can now be published in the current form.

---

## Round 2 · Author Response

Updated version according to referee comments.

You are currently on this page

Resubmission scipost_202102_00019v2 on 23 March 2021

---

## Editorial Decision

publication_decision_taken:_accept